# Semi-ProtoPNet Deep Neural Network for the Classification of Defective Power Grid Distribution Structures

**DOI:** 10.3390/s22134859

**Published:** 2022-06-27

**Authors:** Stefano Frizzo Stefenon, Gurmail Singh, Kin-Choong Yow, Alessandro Cimatti

**Affiliations:** 1Fondazione Bruno Kessler, Via Sommarive 18, 38123 Trento, Italy; cimatti@fbk.eu; 2Department of Mathematics, Informatics and Physical Sciences, University of Udine, Via delle Scienze 206, 33100 Udine, Italy; 3Faculty of Engineering and Applied Science, University of Regina, Wascana Parkway 3737, Regina, SK S4S 0A2, Canada; gurmail.singh@uregina.ca (G.S.); kin-choong.yow@uregina.ca (K.-C.Y.)

**Keywords:** power grid inspection, computer vision, convolutional neural networks, deep learning, insulator classification

## Abstract

Power distribution grids are typically installed outdoors and are exposed to environmental conditions. When contamination accumulates in the structures of the network, there may be shutdowns caused by electrical arcs. To improve the reliability of the network, visual inspections of the electrical power system can be carried out; these inspections can be automated using computer vision techniques based on deep neural networks. Based on this need, this paper proposes the Semi-ProtoPNet deep learning model to classify defective structures in the power distribution networks. The Semi-ProtoPNet deep neural network does not perform convex optimization of its last dense layer to maintain the impact of the negative reasoning process on image classification. The negative reasoning process rejects the incorrect classes of an input image; for this reason, it is possible to carry out an analysis with a low number of images that have different backgrounds, which is one of the challenges of this type of analysis. Semi-ProtoPNet achieves an accuracy of 97.22%, being superior to VGG-13, VGG-16, VGG-19, ResNet-34, ResNet-50, ResNet-152, DenseNet-121, DenseNet-161, DenseNet-201, and also models of the same class such as ProtoPNet, NP-ProtoPNet, Gen-ProtoPNet, and Ps-ProtoPNet.

## 1. Introduction

Electric power grids are responsible for supplying electricity to the consumer with security and reliability. Many distribution networks are installed outdoors without insulation on the conductors; thus, these networks become vulnerable to environmental conditions [1]. A major problem of electrical power systems installed outdoors is the presence of contaminants, which accumulate on the structures and increase the conductivity of the insulating components [2].

With higher surface conductivity, power grid components have a higher leakage current that leads to disruptive discharges [3]. When discharges occur on the surface of the insulation, the contamination burns and becomes encrusted, making cleaning of these components with rain difficult [4]. From components with high encrusted contamination, several discharges occur, which reduce the power quality of the electrical power system [5].

To improve the ability to identify damaged components, inspections are performed on the electrical power system [6]. Inspections are usually carried out from the ground by specialized teams using specific equipment [7], or aerially, usually through images with unmanned aerial vehicles (UAV) [8]. Among the equipment used in the inspections, the ultrasound [9], radio interference [10], ultraviolet camera [11], and infrared camera [12] are highlighted.

Nowadays, advanced image-based models have shown promise for power grid inspections [13]. Specifically, for the image classification task, the state-of-the-art ProtoPNet [14] models stand out. The great advantage of this class of models is that the network finds prototypical parts and combines evidence from the prototypes to make a final classification. Model variations such as NP-Proto-PNet [15], Gen-ProtoPNet [16], Ps-ProtoPNet [17], and Quasi-ProtoPNet [18] are efficient for classification in view of their ability to have interpretability in various applications.

A representative dataset is one of the great difficulties of using deep learning models, as failures are rare in the electrical power system, and it is difficult to have a representative database to train the model. Based on the need to identify failures in a preventive way with a small dataset, this paper proposes the *semi prototypical part network* (Semi-ProtoPNet) for the classification of adverse conditions in distribution network structures. The proposed model is called Semi-ProtoPNet because it does not use all the training steps to avoid a reduction in accuracy. The proposed Semi-ProtoPNet is a non-interpretable model of the ProtoNet class with some advantages that will be explained in this paper. The contributions of this paper to the inspection of the electrical power grid are summarized below:The first contribution is due to the need for a small database to train the proposed Semi-ProtoPNet. Typically, deep neural networks need a large database to train the model. From the proposed method, high accuracy was obtained using a small dataset, which would enable the use of this model for field applications.The proposed model has better accuracy than the state-of-the-art models (VGG-13, VGG-16, VGG-19, ResNet-34, ResNet-50, ResNet-152, DenseNet-121, DenseNet-161, DenseNet-201, ProtoPNet, NP-ProtoPNet, Gen-ProtoPNet, and Ps-ProtoPNet) for image classification. This is because the Semi-ProtoPNet uses a generalized convolutional layer that helps it to use both positive and negative reasoning processes. The idea of using a negative reasoning process is similar to the idea of solving a multiple-choice question, where it becomes helpful to rule out the options that are surely not an answer to the question.The third contribution is related to the use of real inspection images of the electrical power grid. There is great difficulty in obtaining an adequate database to classify the conditions of distribution networks. This occurs because the failures are difficult to find due to the large extension of the network, among other reasons. In this paper, the analysis of adverse conditions is performed based on real inspection images of problematic branches reported by the electric utility.Considering that the proposed model does not focus on a specific condition or component, it has the ability to handle large variations between inspection photos with different image frames, brightness, and backgrounds. This makes inspection easier for the operator, as it is easier to take the photos; therefore, it is a more comprehensive method for this evaluation.

The continuation of this paper is organized as follows: In Section 2, the related works are described and the dataset is presented. In Section 3, the proposed method is presented, and its advantages and differences from previous versions of the ProtoPNet are highlighted. In Section 4, the results are discussed and evaluated. In Section 5, the conclusion is presented.

## 2. Related Works and Considered Dataset

There are several faults that can occur in the electrical power system because it is mostly installed outdoors [19]. One of the causes of failures in the electrical power system is the presence of contamination on the insulating components resulting in greater surface conductivity [20]. With the highest conductivity, the site becomes more susceptible to the development of flashovers, which may result in a shutdown of electricity [21].

Shutdowns in the distribution networks result in a lower level of reliability of the grid, which is a negative result for the electricity utility [22]. Research has been conducted to improve the insulation performance of components in the electrical distribution grid [23]. From the need to have a reliable electrical power system, there is the challenge of automating electrical inspections. Computing [24] and robotics [25] make inspections more dynamic, improving the identification of faults.

Ibrahim et al. [26] presented a study about insulator surface erosion, which is an issue that can be related to the contamination problem; they achieved 89.5% accuracy in the classification of faults. Prates et al. [27] performed an analysis to identify defects in distribution lines, which is the same goal of this paper; using a laboratory-produced dataset, they achieved 85.48% accuracy when identifying defects in insulators. There are also modern failure assessment models that are based on predicting the development of an anomaly regarding the increase in an adverse condition [28].

Monitoring power grids with aerial images is a cost-effective alternative. This solution is becoming increasingly used due to the ability to process large datasets through deep learning neural networks. Models that stand out for this application are VGG, ResNet, and DenseNet among other convolutional neural network (CNN)-based models. In addition to contamination [29], the structures installed outdoors are exposed to freezing, wet, and snowing conditions [30]. Due to these conditions, image analysis using CNNs is a feasible and promising alternative for electrical system inspections.

Many authors have researched the use of aerial images recorded by UAV for electrical power system inspection. Sampedro et al. [31], Tao et al. [32], and Han et al. [33] applied the CNN for fault identification; Miao et al. [34] used a single shot multibox detector (SSD). Another approach that is currently being used is the detection of objects that have faults, such as broken insulators. Techniques such as the region-based convolutional neural network (R-CNN) presented by Li et al. [35] and Li et al. [36], and *you only look once* (YOLO) [37] are being widely used for insulator fault identification. The great advantage of this strategy is that it is possible to identify the exact location of the failure [38]; then, specialized teams can be directed to solve the problem by having its cause and location defined in advance.

Techniques based on deep learning are becoming increasingly popular for the identification of faults in electrical power networks [39]. Fahim et al. [40] proposed the capsule network with sparse filtering to classify faults in transmission lines. The great advantage is that this method does not require a large number of images, which usually is a problem in inspections. Zhao et al. [41] presented an approach based on adaptive parametric linear rectifier units to improve resource learning in deep residual networks for fault diagnosis.

The fusion between wavelet transform and deep residual networks is effective for fault diagnosis, as the vibration resulting from the faults can be evaluated through a series of combined frequency band techniques that result in an improvement of the model [42]. According to Siniosoglou et al. [43], the detection of anomalies using deep learning strategies brings greater reliability in the diagnosis of the network condition.

From the extraction of characteristics using techniques such as an improved AlexNet, it becomes possible to detect anomalies in the electrical power network based on image analysis [44]. To improve the identification of adverse conditions, generating more accurate classifier model techniques such as CNNs based on faster regions are used so that the analysis can be focused on the problem [45].

From advanced models, high accuracy may be obtained to identify faults in the electrical power systems, making it possible to determine the locations that need future maintenance [46]. One of the major difficulties for image analysis in electric power networks is the great diversity of adverse conditions and complex backgrounds, which make the problem difficult to analyze [47]. Furthermore, the number of images that have defective equipment is usually small.

According to Wen et al. [48], the use of a small dataset is a challenge in this analysis. This is mainly because system failures are seldom found during inspections since, when a failure occurs, the electrical utility replaces the components immediately. To address this problem and avoid overfitting, various augmentation techniques can be applied to increase the dataset. Rotation, blurring, scaling, and noise inclusion are techniques that can be used to increase the number of images [37].

To enhance the generalization of the model, it is necessary to have a large number of training samples. With more data for training, it is possible to evaluate all the possible variations of data to have a stable model. Data augmentation is a strategy that can improve the generalization of the method [49]. Techniques such as blur, scaling, and rotation are significantly important to increase the size of data. Based on the problem described in this section, a dataset was created using real images of electrical branches in faulty and good conditions.

### Dataset

The dataset was recorded in southern Brazil according to reports of problems on the branch, given by the local electricity utility *Centrais Elétricas de Santa Catarina*. In this paper, the classification was performed in relation to two classes, defective (first class) and normal (second class). Figure 1 presents some images of the used dataset, available at: https://github.com/SFStefenon/InspectionDataSet (accessed on 17 June 2021).

The photographs recorded during inspections of the electrical power system are taken at different positions, with variations in the image approximation in relation to the network components. The photographs are intended to highlight the conditions of the insulating components, which are responsible for insulating the distribution system and the supporting structure of the grid.

The structures A, B, and C shown in Figure 1 are from a network that has high contamination on its surface. This contamination comes from organic waste that adheres to the structure of the network, a fact that occurs mainly in rural regions. Comparatively, D, E, and F of this figure are photographs taken from networks in good condition that do not have a high concentration of contamination.

In addition to surface contamination of support structures of the power network, during inspections, broken insulators can be found, which is an even greater problem, considering that cleaning the network is not enough to solve this issue. When the insulator has its constructive properties damaged, it is necessary to replace it. To avoid disconnection from the network, this task can be performed by a specialized team with the network energized, having a higher cost when it comes to corrective maintenance [50].

The dataset was built with the purpose of evaluating all the structures surrounding the power grid, such as the crossarms, poles, and insulators. In this way, the presence of strange objects such as bird nests are considered as places that have adverse conditions. This approach was taken because strange objects near the distribution line cause discharge to the ground, which is a problem for the electrical power utility.

The photographs were taken during the inspections performed on equivalent distribution networks in the Santa Catarina State in Brazil, being medium-voltage conventional power grids. In this way, the model can be used in any electric branch that has the same construction features as the site where the inspections were made. This type of structure is present in a major part of the electrical distribution networks in southern Brazil. The model proposed in this paper can be applied to other types of structures, being necessary to perform new inspections to define an updated training dataset for power grids that have a different profile from the one considered in this work.

The dataset used in this paper has 120 images of defective structures and 120 images of structures in good condition, totaling 240 images. All images of the dataset are original and were recorded during inspections of the electrical power grid. Initially, an evaluation of the change in dataset size is presented; after that, all the analyses use the same configuration to compare the best methodology.

The training set has 84 images of defective structures and 84 images of normal structures. The testing set has 36 images of defective structures and 36 images of normal structures. This means that 70% of the data were used for network training and 30% for testing. In this work, the initialization and division of the training, testing, and validation data was performed randomly; no cross-validation was performed. All used images were dimensioned to 224×224, as required by the base models.

## 3. Methodology

In this section, the applied method, and considerations for the analysis are presented. The proposed Semi-ProtoPNet deep learning model stands out for image classification because the prototypes are not latent patches of the training images. They are tensors with values close to values of latent patches of the output of the convolutional layers of the base models.

Along with the positive reasoning process, including the negative reasoning process to reject incorrect classes of an input image, Semi-ProtoPNet does not perform convex optimization of its last dense layer to keep the weights constant. The consideration of both types of reasoning processes improves the performance of the model, making it possible to carry out an analysis with a low number of images that have different backgrounds.

The positive reasoning process means the positive connection between similarity scores of the prototypes and the logits of a correct class; whereas, the negative reasoning process means a negative connection between similarity scores of the prototypes and logits of incorrect classes.

Previous models such as ProtoPNet [14], NP-Proto-PNet [15], Gen-ProtoPNet [16], and Ps-ProtoPNet [17] use prototypes that are latent patches of the training images. The replacement of prototypes with latent patches of training images leads to a decrease in the logit for the correct class of the input image and an increase in the logits of incorrect classes that further leads to a decrease in the accuracy of the ProtoPNet models, see Theorem 2.1 in [14] and Theorem 1 in [17]. The proposed Semi-ProtoPNet outperforms these applications, as will be presented here.

The replacement of prototypes with latent patches reduces the accuracy because there can be only few images that have identical patches, but some pixel values of two patches can be close to each other. Then, similarity of an input image with prototypes of its own class can be reduced if we use prototypes as latent patches of the training image, which further lead to a decrease in the accuracy. For this reason, the nonreplacement of prototypes with latent patches and the use of both types of reasoning processes (positive and negative) helps the model perform classifications successfully even with the small datasets.

### 3.1. Architecture

The ProtoPNet class classifies the images on the basis of a weighted combination of the similarity scores of the latent patches of the training images [14]. For each class (normal and defective), a fixed number of prototypes (which are replaced with the latent patches) are selected. For this application, 10 prototypes for each class were used. Based on this topology, the Semi-ProtoPNet structure is defined.

Figure 2 presents the architecture of the Semi-ProtoPNet with VGG-19 [51] as the baseline, though Semi-ProtoPNet can also be constructed over the convolutional layers of some other base models. After the convolution layers, there are two additional layers of dimensions 2×2 and 1×1, respectively. These convolutional layers *ℓ* are followed by a generalized convolution layer of prototypical parts $p_{p}$ [52] and a dense layer *w* with weight matrix $m_{w}$.

The rectified linear unit (ReLU) activation function [53] is used for the first additional convolutional layer and the sigmoid activation function [54] is used for the second additional convolutional layer. The use of ReLU and sigmoid activation functions are the most appropriate for CNN-based image classification of this class of algorithms, as used in [14,16,18] for models of the same class. For an input image *x*, ℓ(x) is the output of *ℓ*, where the shape of ℓ(x) is 512×6×6. Thus, $P^{k}={\left\{p_{l}^{k}\right\}}_{l=1}^{m^{\prime }}$ is a set of prototypes of class *k* and $P={\left\{P^{k}\right\}}_{k=1}^{n}$ is the set of prototypes of all classes, $m^{\prime }$ is the number of prototypes for each class, and *n* is the total number of classes. In this approach, $m^{\prime }=10$, n=2, and the hyperparameter $m^{\prime }=10$ is chosen randomly.

The shape of each prototype is 512×h×w, where 1×1<h×w<6×6. Therefore, every prototype can be considered a representation of some prototypical part of the image. Semi-ProtoPNet calculates the similarity scores between an input image and the prototypical parts $p_{1}^{1}-p_{10}^{1}$ and $p_{1}^{2}-p_{10}^{2}$. In layer *w*, the matrix *S* is multiplied with $m_{w}$ to obtain the logits. To achieve a complete analysis of the model structure, the architecture of the proposed Semi-ProtoPNet with VGG-19 is compared using the several VGG, ResNet, and DenseNet baselines. At the end of this paper, the proposed model is compared with the baselines and models of ProtoPNet class.

The acronym VGG refers to Visual Geometry Group; it is a standard multilayer deep CNN architecture. Deep refers to the number of layers, e.g., VGG-13 [55], VGG-16 [56], and VGG-19 [57], have 13, 16, and 19 convolutional layers, respectively. These models are structured as a series of convolutional layers, which can efficiently extract features from the data. After the first convolutional layers, max-pooling layers are used to compute the maximum of a local patch of units in a feature map. At the end of the model, a fully connected layer is used to perform the classification [58].

A residual neural network (ResNet) is one of the first artificial neural networks used in deep learning using hundreds of layers; one of the great advantages of this architecture is that it maintains good performance results even when using a large number of layers, being possible to compute many more layers than previous models [59]. Given the universal approximation theorem, a feedforward network with a single layer is enough to represent any given function. Nevertheless, the layer can be massive, and the net is likely to overfill the data. Thus, there is a general trend in the research community to use deeper architectures, making the ResNet model promising. Similar to VGG networks, the ResNet model has several variations, such as ResNet-34 [60], ResNet-50 [61], and ResNet-152 [62], which depend on the number of layers used.

The dense convolutional network (DenseNet) is a type of CNN that uses dense connections between layers, where all layers are connected directly to each other. In each layer, the feature maps of all previous layers are used as inputs and their feature maps are utilized as inputs in all following layers [63]. The major advantages of using DenseNets are that they alleviate the vanishing gradient problem, strengthen feature propagation, and encourage feature reuse, thus reducing the number of parameters in the network and making it more efficient. This structure also has variations according to the number of its layers, such as DenseNet-121 [64], DenseNet-161 [65], and DenseNet-201 [66].

The NP-ProtoPNet attempts to simulate human reasoning for image recognition while comparing the parts of a test image with the corresponding parts of known class images [15]. The accuracy of NP-ProtoPNet achieves values comparable with the best non-interpretable deep learning models. While the ProtoPNet and NP-ProtoPNet use prototypes of spatial dimension 1×1 and the distance function L2, the Gen-ProtoPNet [16] uses a generalized form of the distance function, which allows the use of prototypes of any spatial dimension. According to Singh and Yow [17], the Ps-ProtoPNet classifies images by recognizing objects rather than the background in the images. Quasi-ProtoPNet is an interpretable model that considers only the positive reasoning process [18].

#### Training Procedure

The generalized distance function *d* of the Euclidean distance function L2 was used in the Semi-ProtoPNet. The shape of ℓ(x) is 512×6×6, where 512 is the depth of ℓ(x) and 6×6 are its spatial dimensions. The output *z* of the convolutional layers *ℓ* has (7−h)(7−w) patches of dimensions h×w. The square of the distance $d(Z_{ij},p)$ between the prototype *p* and the patch $Z_{ij}$ of *z* is given by
(1)$$d^{2}\left(Z_{ij},p\right)=\sum_{l=1}^{h}\sum_{m=1}^{w}\sum_{k=1}^{512}\left|\right|z_{(i+l-1)(j+m-1)k}-p_{lmk}{\left|\right|}_{2}^{2}.$$
where $p_{lmk}$ denotes a *k*th prototype of length *l* and width *m*.

For prototypes of spatial dimension 1×1, where h=w=1, the square of the Euclidean distance between the prototype *p* and a patch of *z* is
(2)$$d^{2}\left(Z_{ij},p\right)=\sum_{k=1}^{512}\left|\right|z_{ijk}-p_{11k}{\left|\right|}_{2}^{2},$$
where $p_{11k}≃p_{k}$. Therefore, the distance function *d* is a generalization of L2. Then, $p_{p}$ is calculated according to
(3)$$p_{p}\left(z\right)=\max_{Z\in \mathrm{patches}(z)}\log\left(\frac{d^{2}\left(Z,p\right)+1}{d^{2}\left(Z,p\right)+\epsilon }\right).$$

Equation (Equation 3) shows that a prototype is more similar to the input image if the inverse of the distance between a latent patch of the image and the prototype is smaller.

In the proposed Semi-ProtoPNet, all layers are optimized before the dense layer. Considering that $X=\{x_{1}\ldots x_{n}\}$ and $Y=\{y_{1}\ldots y_{n}\}$ are, respectively, sets of images and corresponding labels, where $D=\{\left(x_{i},y_{i}\right):x_{i}\in X,y_{i}\in Y\}$, the objective function to be optimized is
(4)$$\min_{P,\ell_{conv}}\frac{1}{n}\sum_{i=1}^{n}\mathrm{CrosEnt}\left(h\circ p_{p}\circ \ell \left(x_{i}\right),y_{i}\right)+\lambda_{1}\mathrm{ClstCst}+\lambda_{2}\mathrm{SepCst},$$
where cluster cost (ClstCst) and separation cost (SepCst) are
(5)$$\mathrm{ClstCst}=\frac{1}{n}\sum_{i=1}^{n}\min_{j:p_{j}\in P_{y_{i}}}\min_{Z\in \mathrm{patches}(\ell \left(x_{i}\right))}d^{2}\left(Z,p_{j}\right);$$
(6)$$\mathrm{SepCst}=-\frac{1}{n}\sum_{i=1}^{n}\min_{j:p_{j}\notin P_{y_{i}}}\min_{Z\in \mathrm{patches}(\ell \left(x_{i}\right))}d^{2}\left(Z,p_{j}\right).$$

According to Equation (Equation 5), the decrease in the ClstCst leads to the grouping of prototypes around their classes. However, based on Equation (Equation 6), the decrease in SepCst keeps prototypes away from their incorrect classes. Finally, Equation (Equation 4) shows that the drop in cross-entropy leads to improvement in the classification. As the distance function is non-negative, optimizing all layers except the last layer with the stochastic gradient descent (SGD) optimizer [67] helps Semi-ProtoPNet to learn important latent space.

Observing that $m_{w}$ is the weight matrix for the last layer, $m_{w}^{\left(i,j\right)}$ is the weight assigned to the connection between the similarity score of *j*th prototype and logit of *i*th class; in a class *k*, $m_{w}^{\left(i,j\right)}=1$ is defined for all *j* with $p_{j}^{i}\in P^{i}$, and for all $p_{j}^{k}\notin P^{i}$ with k≠i, $m_{w}^{\left(k,j\right)}$ equal to −1, where $\lambda_{1}$ and $\lambda_{2}$ are hyperparameters belonging to {0.7,0.8}. Therefore, the weight matrix is given by
(7)$$m_{w}=\left[\begin{array}{cccccc} 1 & \ldots & 1 & -1 & \ldots & -1\\ -1 & \ldots & -1 & 1 & \ldots & 1 \end{array}\right]$$

In the proposed method, the convex optimization of the last layer is not performed to keep the impact of negative reasoning in the image classifications process. The SGD optimizer updates the parameters to minimize the loss function [68], taking steps at each iteration towards the negative loss gradient,
(8)$$\theta_{i+1}=\theta_{i}-\alpha \nabla F\left(\theta_{i}\right)$$
where θ is the vector to be minimized, α is the learning rate, and F(θ) is the loss function.

The computational effort in the training phase of the proposed model is higher in relation to standard neural networks. As the training of the model is done offline, reducing the time to train is not the objective of this methodology, as the goal of the proposed model is to achieve as high an accuracy as possible for the classification task. Considering that the computational effort of testing is considerably low, after the training phase, testing of the conditions can be performed in the field with embedded systems.

Compared with previous versions of ProtoPNet, the proposed Semi-ProtoPNet has the following advantages:The proposed method does not replace prototypes with the latent patches of the training images; these prototypes have values very close to the pixel values of the training images.The prototypes with spatial dimensions bigger than 1×1 are used. With the generalized distance function *d*, it is possible to use prototypes with any type of spatial dimensions—that is, square spatial dimensions as well as rectangular spatial dimensions.The Semi-ProtoPNet does not perform convex optimization of the last layer to maintain the impact of the negative reasoning process on the image classification, whereas the ProtoPNet model emphasizes the positive reasoning process. Further, the nonoptimization of the last layer reduces the training time considerably.Using the Semi-ProtoPNet, regardless of the weight given to the positive class, it gives exactly equal to the negative of that weight to the negative class, and this weight is not reduced to zero, unlike ProtoPNet. By doing so, we equally consider both positive reasoning and negative reasoning to classify the images.

### 3.2. Limitations

Theorem 2.1 in [14] and Theorem 1 in [17] provide lower bound in the decrease in logit for correct class and increase in the logits of incorrect classes when prototypes are replaced with the latent patches of input images. So, if the change in the logits of the other ProtoPNet models is far from the bounds provided by theorems, then Semi-ProtoPNet may not perform better than the other ProtoPNet models.

### 3.3. Performance Evaluation Metrics

For comparison purposes, the accuracy Equation (Equation 9), precision Equation (Equation 10), recall Equation (Equation 11), and F1-score Equation (Equation 12) measures were evaluated, given by
(9)$$\mathrm{Accuracy}=\frac{TP+TN}{\mathrm{Total}\mathrm{Cases}},$$
(10)$$\mathrm{Precision}=\frac{TP}{TP+FP},$$
(11)$$\mathrm{Recall}=\frac{TP}{TP+FN},$$
(12)$$F1-\mathrm{score}=\frac{2}{{\mathrm{Precision}}^{-1}+{\mathrm{Recall}}^{-1}},$$
where abbreviations refer to true positive (*TP*), true negative (*TN*), false positive (*FP*), and false negative (*FN*). In the confusion matrices, the values are presented in relation to the defective and normal classes, and each matrix corresponds to the evaluation of a different model. For the final comparative analysis, the test of hypothesis, standard deviation, and kurtosis were calculated.

Since accuracy is the proportion of correctly classified images among all the test images, the test of hypothesis concerning a system of two proportions is applied. If the size of test dataset *n* and the number of images correctly classified by models 1 and 2 are $x_{1}$ and $x_{2}$, respectively, then ${\tilde{p}}_{1}=x_{1}/n$ and ${\tilde{p}}_{2}=x_{2}/n$. The statistic for the test concerning the difference between two proportions is
(13)$$Z=\frac{{\tilde{p}}_{1}-{\tilde{p}}_{2}}{\sqrt{2\tilde{p}\left(1-\tilde{p}\right)/n}},$$
where ${\tilde{p}}_{1}$ and ${\tilde{p}}_{2}$ are the accuracies given by the compared methods, and $\tilde{p}$ is calculated by
(14)$$\tilde{p}=\left(x_{1}+x_{2}\right)/2n.$$

Therefore, the hypothesis is as follows:(15)$$\left\{\begin{array}{c} H_{0}:\left(p_{1}-p_{2}\right)=0\;\;\;\left(\mathrm{null}\;\mathrm{hypothesis}\right),\\ H_{a}:\left(p_{1}-p_{2}\right)\neq 0\;\;\;\left(\mathrm{alternative}\;\mathrm{hypothesis}\right). \end{array}\right.$$

The test of hypothesis was performed for the level of confidence (α) = 0.01. As the hypothesis is two-tailed, the *p*-value must be less than 0.005 to reject the null hypothesis. In this hypotheses test, $p_{1}$ is the accuracy given by Semi-ProtoPNet and $p_{2}$ represents the accuracies given by the other models. The values of test statistic *Z* are given by Equation (Equation 13).

The simulations were performed in a Deep Learning Server *(Lambda Labs of the University of Regina, Canada)*; the specifications of this cluster are presented in Table 1. The algorithm proposed in this paper was developed in Python.

The flowchart of the procedure performed for this research is presented in Figure 3. The development of this project began with field inspections carried out by a specialized team after the indication that the evaluated distribution branch had high evidence of faults with disconnection due to the presence of contamination. The inspections were conducted in the state of Santa Catarina, in southern Brazil.

## 4. Results and Discussion

In this section, the results of the proposed method are presented and discussed. To have a global analysis, the evaluation will be presented with different base models to define the best structure of the proposed method. Then, the proposed model will be compared with these state-of-the-art models using the base model by itself.

The first evaluation is performed in relation to the dataset, with the goal of verifying the influence of changing the size of the used dataset on the model’s performance. The comments and evaluation are related to accuracy and F1-score. The best results of each model are underlined and the best overall result is shown in bold.

### 4.1. Dataset Evaluation

The dataset can be a limiting factor in the use of deep layer models due to the need for a large number of images to perform the training. For this reason, a reduction in the number of images is evaluated. The VGG, ResNet, and DenseNet class models are evaluated using the database reduced from 240 to 160 images, maintaining a balanced distribution between “damaged” power grids (80 images) and networks in good condition (80 images). The results of these variations are presented in Table 2.

The reduction of the database is a major issue; as can be seen in Table 2, all the models evaluated had a lower F1-score and most of them had a lower accuracy using a smaller database. This further highlights the difficulty in performing the analysis with a reduced number of images, which is the goal of the model proposed in this paper.

Comparatively, Sampedro et al. [31], Jiang et al. [46], Zhang et al. [69], and Tao et al. [32], respectively, used 160, 385, 400, and 600 images to identify adverse conditions on the grid. All these authors highlight the difficulty in dealing with small datasets. Following the analysis, considering that the dataset is sufficient to obtain reasonable accuracy and F1-score results, the complete evaluation of the proposed model is presented here.

### 4.2. Confusion Matrices

The confusion matrices of Semi-ProtoPNet with different base models are presented in Figure 4. From the confusion matrices, the accuracy, precision, recall, and F1-score are obtained. These results are used to compare the performance of the structure of the model using different baselines. From the baseline change, the structure is also updated, thus generating a variation of the model.

### 4.3. Baseline Evaluation

The first evaluation of the structure of the proposed Semi-ProtoPNet (SPPN) is the use of different baselines. The results of this variation are presented in Table 3. Using VGG-19 as a baseline, the results of Semi-ProtoPNet were considerably promising for field applications, considering that the accuracy of 97.22% and a F1-score of 0.9729 were achieved.

The ResNet as a base model results in inferior performance regarding the evaluated metrics, the best accuracy and F1-score were obtained with ResNet-50 being inferior to the previously analyzed VGG-19. Using DenseNet, the results were also inferior to VGG-19; based on this, VGG-19 is defined as the standard baseline.

These results prove that sometimes the use of more layers in the structure of the deep neural network is not a good strategy, as this could require more computational effort and does not improve the performance of the model. This shows that to have an optimized structure, it is important to evaluate several variations of the parameters.

The Semi-ProtoPNet has acceptable results, even changing the baseline, showing that it is not the baseline that makes the method reach high result values, it is the proposed method by itself. This statement can also be made when analyzing the difference between the convergence of the Semi-ProtoPNet to different baselines; the results of these comparisons are presented in Figure 5.

As can be seen, all variations of the model achieve convergence in less than 20 epochs with a stable result. All analyses were carried out until 100 epochs, so there was certainty about the convergence of the algorithm and its stability.

### 4.4. Benchmarking

Table 4 presents a comparison of the proposed method with the VGG, ResNet, and DenseNet class algorithms and the family of ProtoPNet models. This comparison aims to assess whether the result occurs because the proposed method is superior or if it happens in other equivalent models. For a fair analysis, the ProtoPNet models use the VGG-19 baseline, which was the best backbone previously found.

The Semi-ProtoPNet with VGG-19 (SPPN-VGG-19) presented in this paper has better results than all variations of the compared ProtoPNet models. The presented results highlight that even models for this specific task have lower results than the proposed SPPN-VGG-19. This probably occurs because of the small number of images, which is a common problem in the inspections. This proves that the SPPN-VGG-19 is well-indicated for this evaluation.

The nonreplacement of prototypes with the patches of the training images, the nonoptimization of the last layer, and the use of prototypes with rectangular spatial dimensions and square spatial dimensions greater than 1×1 helped the proposed model to improve its performance.

#### 4.4.1. Test of Hypothesis for the Accuracy and Statistical Evaluation

As mentioned in Section 3.3, the test of hypothesis concerning a system of two proportions is applied to see whether the accuracy given by the proposed model is statistically significantly better than the accuracies given by the other models. Considering that α is 0.01, the null hypothesis for all the *p*-values listed in Table 5 are rejected. Based on the value of the α this analysis has 99% confidence that the accuracies given by SPPN-VGG-19 are significantly better than the accuracies given by each of the other models.

The result of the statistical Z-statistic and *p*-value are not presented for SPPN-VGG-19, as this model is used for comparison with other models. The statistical results show that there is stability in the compared models, although some models result in low accuracy.

#### 4.4.2. State-of-the-Art Approaches

SPPN-VGG-19 outperformed other authors who used CNNs for the equivalent purpose. Han et al. [33] achieved a F1-score of 0.9466 and Tao et al. [32] achieved 0.9340 using CNNs for insulator faults detection.

Liu et al. [37] had a F1-score of 0.9499 using YOLOv3 and Feng et al. [70] had a F1-score of 0.9293 using YOLOv5, which is the most current model for object detection nowadays. The applications of Jiang et al. [46] and Miao et al. [34] using single shot multibox detector had a F1-score of 0.9244 and 0.9184, respectively. In this paper, using SPPN-VGG-19, a F1-score of 0.9729 was reached in the evaluation of power grids structures with adverse conditions. This method also proved that it is stable when statistical analysis is evaluated.

Comparison with previous works showed that the proposed model has a better F1-score than other approaches, in addition to the fact that in some studies such as [71], the analysis is performed only with the focus on identifying the chain of insulators and not on the defect classification, which is necessary for inspections of the electrical power system. The proposed method presented in this paper was applied to evaluate the entire structure of the grid and if an adverse condition is present near the insulators.

## 5. Conclusions

The identification of failures in the distribution networks improves the quality of the electric energy supply since it is possible to determine preventive maintenance strategies to correct failures before network outages occur. Contamination is a problem found in several networks that are close to unpaved streets, especially in rural areas. As soon as the contamination becomes encrusted, it is necessary to perform the maintenance of the network to ensure its operation. This paper proves that using deep learning for computer vision is possible to classify adverse conditions on the network, considering that through the proposed model acceptable values were reached to use the model in field applications.

The use of the proposed Semi-ProtoPNet model showed considerable promise for the analysis in question, considering that the accuracy of 97.22% was obtained for the classification of adverse conditions in distribution networks. Using VGG-19 as the baseline, the proposed method was superior to models of the same class such as ProtoPNet, NP-ProtoPNet, Gen-ProtoPNet, and Ps-ProtoPNet, in addition to being superior to the analysis carried out by other authors for equivalent problems.

As was presented, changing the structure of the network results in variation in its performance; so, it is necessary to carry out an analysis using several baselines in order to obtain the best structure of the model. The result of this work was promising, since the dataset used is based on real images without preprocessing, where there are great variations in image background, brightness, and framing, conditions commonly found in photographs of the electrical power grid. This shows that the application has the possibility of being carried out directly for field inspections.

Several authors only evaluate the location of the insulator chain and not specifically network faults. In addition, it is common to use artificial datasets, in which failures were obtained from overlapping, which often does not correspond to the real problems of the electrical power system. For this reason, this paper stands out among all the work already done. Future work can be done by combining the Semi-ProtoPNet, presented in this paper, with specific equipment for the inspection of the network. Cross-validation can be used to improve the generalizability of the evaluation.

## Figures and Tables

**Figure 1 sensors-22-04859-f001:**
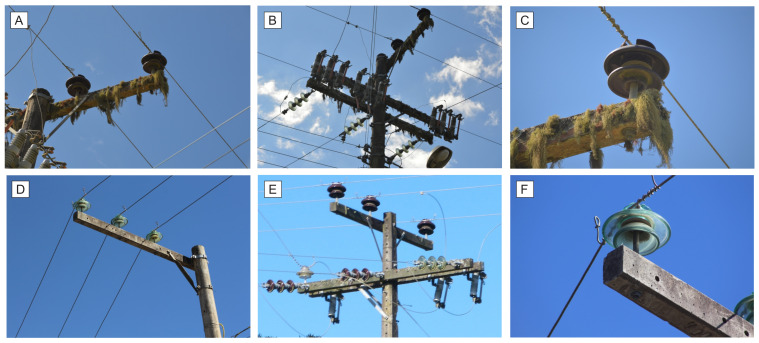
Images of inspections of electric power distribution network.

**Figure 2 sensors-22-04859-f002:**
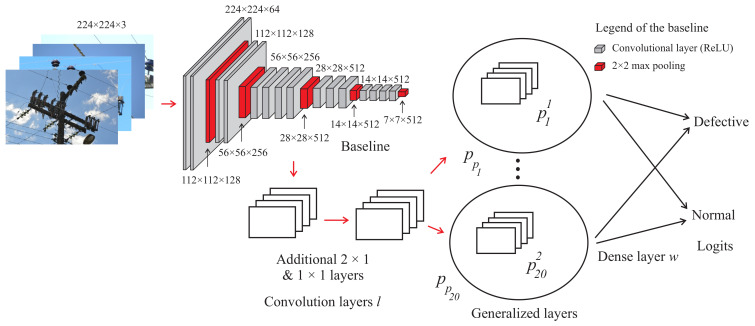
Semi-ProtoPNet architecture.

**Figure 3 sensors-22-04859-f003:**
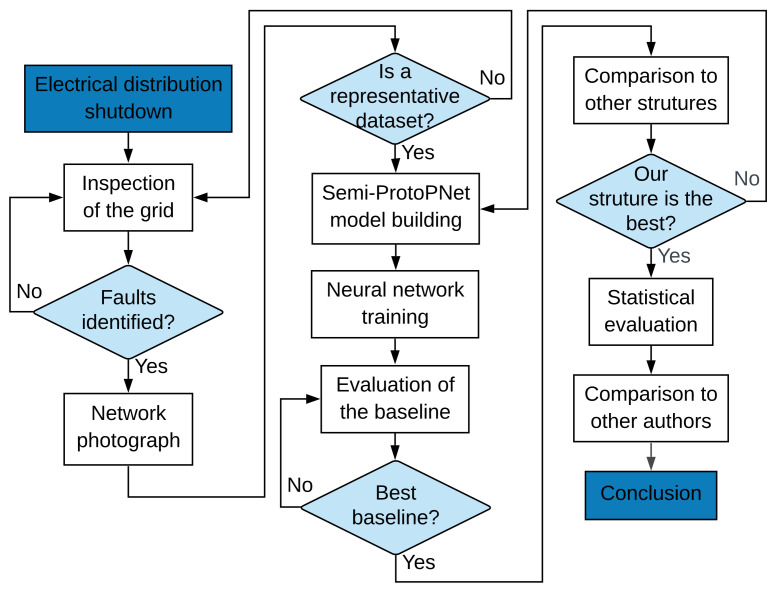
Flowchart of the analysis performed in this paper.

**Figure 4 sensors-22-04859-f004:**
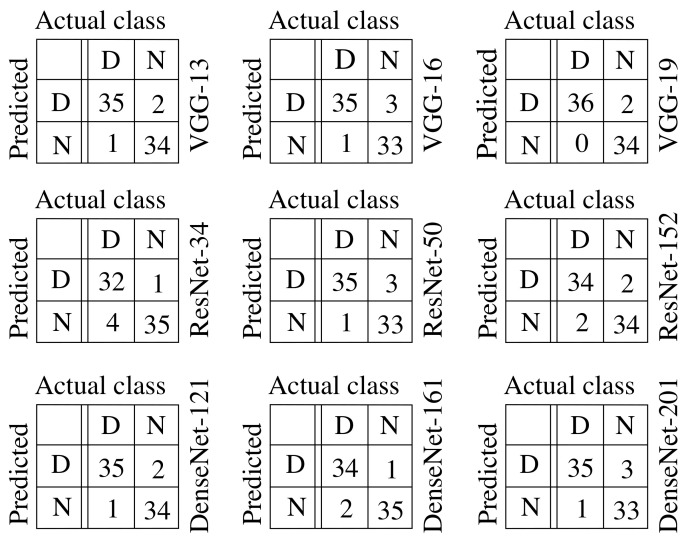
Semi-ProtoPNet confusion matrices with different base models.

**Figure 5 sensors-22-04859-f005:**
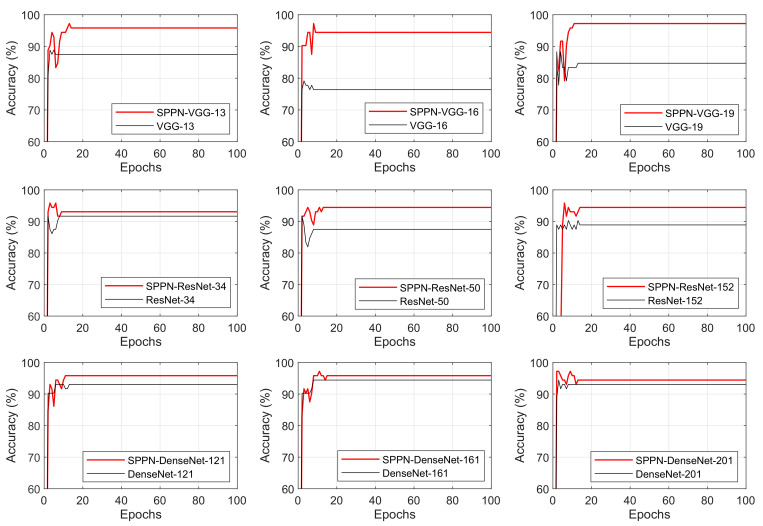
Convergence of Semi-ProtoPNet compared with different baselines.

**Table 1 sensors-22-04859-t001:** Specifications of the used Deep Learning Server.

Description	Specification
Intel(R) Xeon(R) Silver 4214	2.20 GHz
NVIDIA Quadro RTX 5000	8 × GPUs of 16 GB
Random Access Memory	256 GB
Hard Drive (SSD)	1.9 TB

**Table 2 sensors-22-04859-t002:** Assessment of the database reduction.

Model	Numb. Imag.	Accur. (%)	Precision	Recall	F1-Score
VGG-13	240	**91.67**	0.9310	0.8999	0.9153
	160	78.12	0.9999	0.5625	0.7199
VGG-16	240	76.67	0.7222	0.8667	0.7879
	160	81.25	0.9999	0.6250	0.7692
VGG-19	240	76.67	0.7222	0.8667	0.7879
	160	75.00	0.8333	0.6249	0.7143
ResNet-34	240	90.00	0.8529	0.9667	0.9062
	160	90.62	0.9999	0.8125	0.8965
ResNet-50	240	90.00	0.8529	0.9667	0.9062
	160	90.00	0.8749	0.9333	0.9032
ResNet-152	240	**91.67**	0.8788	0.9667	**0.9206**
	160	87.50	0.9286	0.8125	0.8667
DenseNet-121	240	90.00	0.8999	0.8999	0.8999
	160	84.37	0.9999	0.6875	0.8148
DenseNet-161	240	90.00	0.9285	0.8667	0.8968
	160	87.50	0.9999	0.7499	0.8571
DenseNet-201	240	**91.67**	0.8788	0.9667	0.9206
	160	84.37	0.9999	0.6875	0.8148

**Table 3 sensors-22-04859-t003:** Evaluation of the Semi-ProtoPNet with different baselines.

Baseline for Semi-ProtoPNet	Accuracy (%)	Precision	Recall	F1-Score
SPPN-VGG-13	95.83	0.9459	0.9722	0.9589
SPPN-VGG-16	94.44	0.9210	0.9722	0.9459
SPPN-VGG-19	**97.22**	0.9473	0.9999	**0.9729**
SPPN-ResNet-34	93.05	0.9189	0.9444	0.9315
SPPN-ResNet-50	94.44	0.9210	0.9722	0.9459
SPPN-ResNet-152	94.44	0.9444	0.9444	0.9444
SPPN-DenseNet-121	95.83	0.9459	0.9722	0.9589
SPPN-DenseNet-161	95.83	0.9459	0.9722	0.9589
SPPN-DenseNet-201	94.44	0.9210	0.9722	0.9459

**Table 4 sensors-22-04859-t004:** Benchmarking evaluation.

Evaluated Method	Accuracy (%)	Precision	Recall	F1-Score
VGG-13	88.88	0.9117	0.8611	0.8857
VGG-16	79.16	0.8181	0.7499	0.7826
VGG-19	84.72	0.8205	0.8888	0.8533
ResNet-34	91.66	0.9166	0.9166	0.9166
ResNet-50	91.66	0.9687	0.8611	0.9117
ResNet-152	90.27	0.8536	0.9722	0.9090
DenseNet-121	93.05	0.9189	0.9444	0.9315
DenseNet-161	94.44	0.9210	0.9722	0.9459
DenseNet-201	93.05	0.9428	0.9166	0.9295
ProtoPNet	83.33	0.7999	0.8888	0.8421
NP-ProtoPNet	75.00	0.7249	0.8055	0.7631
Gen-ProtoPNet	88.88	0.8499	0.9444	0.8947
Ps-ProtoPNet	93.05	0.9189	0.9444	0.9315
SPPN-VGG-19	**97.22**	0.9473	0.9999	**0.9729**

**Table 5 sensors-22-04859-t005:** Statistical results for the evaluated models.

Evaluated Method	Z- Statistic	*p*-Value	Standard Deviation	Kurtosis
VGG-13	1.97	2.50 × $10^{-2}$	7.38 × $10^{-1}$	8.01 × $10^{1}$
VGG-16	3.36	4.00 × $10^{-4}$	3.86 × $10^{-1}$	2.81 × $10^{1}$
VGG-19	2.62	4.53 × $10^{-3}$	1.45	2.46 × $10^{1}$
ResNet-34	1.45	7.35 × $10^{-2}$	9.89 × $10^{-1}$	1.67 × $10^{1}$
ResNet-50	1.45	7.35 × $10^{-2}$	8.80 × $10^{-1}$	2.50 × $10^{1}$
ResNet-152	1.72	4.27 × $10^{-2}$	6.69 × $10^{-1}$	4.87 × $10^{1}$
DenseNet-121	1.16	1.23 × $10^{-1}$	6.33 × $10^{-1}$	1.28 × $10^{1}$
DenseNet-161	0.83	2.03 × $10^{-1}$	1.02	1.06 × $10^{1}$
DenseNet-201	1.16	1.23 × $10^{-1}$	6.32 × $10^{-1}$	3.45 × $10^{1}$
ProtoPNet	2.81	2.48 × $10^{-3}$	3.12 × $10^{-2}$	6.17 × $10^{1}$
NP-ProtoPNet	3.85	6.00 × $10^{-5}$	1.25 × $10^{-1}$	4.77
Gen-ProtoPNet	1.97	2.50 × $10^{-2}$	1.42 × $10^{-2}$	1.56 × $10^{1}$
Ps-ProtoPNet	1.16	1.23 × $10^{-1}$	1.71 × $10^{-2}$	2.29 × $10^{1}$
SPPN-VGG-19	-	-	2.99 × $10^{-2}$	2.20 × $10^{1}$

## Data Availability

The data to support the results presented in this paper are available at https://github.com/SFStefenon/InspectionDataSet (accessed on 17 June 2021).

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
