# Peer review of "Semi-ProtoPNet Deep Neural Network for the Classification of Defective Power Grid Distribution Structures"

_sensors, 2022, doi:10.3390/s22134859_

Round 1

Reviewer 1 Report

The paper is well written, it presents results using Semi-ProtoPNet for image classification. However, there are several that should be addressed. My specific comments are the following:

1. The abstract contains many acronyms which lack context. This makes reading of the abstract difficult.

2. Section 2 up to line 158 can be restructured into the Introduction section for better readability.

3. Based on what consideration were the parameters and the structure of the Semi-ProtoPNet selected (p6., lines 247-252, p. 7, lines 279-283)? How sensitive is the overall performance to these selections?

4. What are the limitations and drawbacks of the proposed algorithm? The paper does not discuss these aspects.

5. The conclusion is too long, it should summarize the contributions of the paper.

6. The UAV aspects should only briefly be mentioned in the whole paper as that part has not been investigated yet.

I recommend rewriting the paper based on these comments.

Author Response

Please find the Response to Reviewers attached.

Thank you.

Reviewer 2 Report

Paper can be published in its current form.

Author Response

Thank you for your time to read our work, we appreciate your indication for
the paper to be accepted.

Reviewer 3 Report

1. The references in semi-proto PNet deep neural network for the classification of all power grid distribution structures should be cited for sensors Journal.

2. The differences between  plmk  in equation (1) and p1, p2 in equation (13) should be explained in details.

3. The AMD processor with 8×GPUs with 16 GB of memory  should be listed specifications.

4. The structures for VGG-13, 67 VGG-16, VGG-19, ResNet-34, ResNet-50, ResNet-152, DenseNet-121, DenseNet-161, 68 DenseNet-201, ProtoPNet, NP-ProtoPNet, Gen-ProtoPNet, and Ps-ProtoPNet should be explained.

Author Response

(The authors gave the same response as above.)

Reviewer 4 Report

This paper proposes a classification of contaminated/defective electrical distribution structures by processing a limited dataset in a semi prototypical part deep neural network, which is a variant of the ProtoPNet model. The authors claim better accuracy than similar models and the best adaptability to handle adverse conditions during the capture of the images as different backgrounds and brightness levels.

In the opinion of this reviewer, the paper presents a good novelty grade, but a grammar check must be performed; for instance, there are many repeated phrases/explanations. Also, revise lines 6-10.

Why was cross-validation not performed?

What are the disadvantages of your proposal concerning others?

Author Response

(The authors gave the same response as above.)

Round 2

Reviewer 1 Report

The authors corrected the paper based on my comments for the initial draft. I recommend accepting it as is.